# Dental complexity and diet in amniotes: A meta-analysis

**Anessa C. DeMers** [1] *, **John P. Hunter** [2]

**1** Department of Evolution, Ecology, and Organismal Biology, The Ohio State University, Columbus, Ohio, United States of America, **2** Department of Evolution, Ecology, and Organismal Biology, The Ohio State University, Newark, Ohio, United States of America

* demers.29@osu.edu

## Abstract

Tooth morphology is among the most well-studied indicators of ecology. For decades, researchers have examined the gross morphology and wear patterns of teeth as indicators of diet, and recent advances in scanning and computer analysis have allowed the development of new and more quantitative measures of tooth morphology. One of the most popular of these new methods is orientation patch count (OPC). OPC, a measure of surface complexity, was originally developed to distinguish the more complex tooth crowns of herbivores from the less complex tooth crowns of faunivores. OPC and a similar method derived from it, orientation patch count rotated (OPCR), have become commonplace in analyses of both modern and fossil amniote dietary ecology. The widespread use of these techniques makes it possible to now re-assess the utility of OPC and OPCR. Here, we undertake a comprehensive review of OPC(R) and diet and perform a meta-analysis to determine the overall difference in complexity between herbivores and faunivores. We find that the relationship between faunivore and herbivore OPC or OPCR values differs substantially across studies, and although some support the initial assessment of greater complexity in herbivores, others do not. Our meta-analysis does not support an overall pattern of greater complexity in herbivores than faunivores across terrestrial amniotes. It appears that the relationship of OPC or OPCR to diet is taxon-specific and dependent on the type of faunivory of the group in question, with insectivores often having values similar to herbivores. We suggest extreme caution in comparing OPC and OPCR values across studies and offer suggestions for how OPCR can constructively be used in future research.

## Introduction

Dietary inference based on morphology has been a field of increasing interest over the past two decades. Although it has long been understood that the morphology of animals reflects their ecology, it is only in recent years that technology has progressed to the point that robust, quantitative, and large-scale studies linking morphology to ecology have become possible [1–5]. One method that has frequently been used in such studies is orientation patch count [3]. Orientation patch count (OPC), or a derived version of the method called orientation patch count rotated (OPCR) [6], is used to quantify the complexity of a surface, and was developed

**Data Availability Statement:** All relevant data are within the paper and its Supporting Information files.

**Funding:** The work of ACD was funded through the Distinguished University Fellowship provided by The Ohio State University. The funders had no role

in study design, data collection and analysis, decision to publish, or preparation of the manuscript.

**Competing interests:** The authors have declared that no competing interests exist.

for use on mammal dentitions to distinguish herbivores from faunivores. Thanks to the method's ease of implementation both on its own and in combination with other dental topography metrics, orientation patch count has seen wide application in studies of mammals and other terrestrial amniotes [e.g 3, 7–10]. The widespread adoption seen in the past decade makes it possible for us now to reassess the method's efficacy in distinguishing diets and offer suggestions as to its most appropriate applications.

Orientation patch count was first developed by Evans et al. [3]. The original technique involved subdividing tooth surfaces into "patches" of similar orientation and a specified minimum area. It was used to demonstrate that in both extant rodents and extant carnivorans, the teeth of herbivores have more complex surfaces (more patches) than the teeth of faunivores. This pattern was ascribed by Evans et al. to the differing digestion requirements of plant and animal matter. Plant matter is characterized by tough cell walls and low-digestibility proteins that must be heavily masticated or otherwise processed to extract nutrition, whereas animal matter lacks cell walls and, although it may need to be sliced into smaller units for swallowing, is generally more digestible [11, 12]. Herbivores, then, must extensively process their food in a way faunivores do not [13–15], and Evans et al. reasoned that as such the teeth of herbivores will be divided into more patches to function as "tools" for food processing. Although OPC showed enough promise to be adopted by other researchers [7, 16–22], it is sensitive to changes in the orientation of the tooth row, decreasing the replicability of the results. To correct for this issue, Evans and Jernvall [6] described a new method called orientation patch count rotated. In OPCR, the calculation of patch number is repeated several times with the tooth row at slightly varying orientations.

Since their inception, OPC and OPCR (collectively referred to here as OPC(R)) have been applied to several types of problem. One is the tracking of complexity or morphological disparity in a lineage over time [e.g. 22–30]. Another is to provide measures of similarity among extinct and/or extant groups to estimate functional overlap [e.g. 7, 16, 18, 31–35]. The strength of both of these approaches is that they detect broad-scale trends in complexity rather than attempting to assign meaning to individual OPC(R) values. Commonly, however, OPC(R) is used to infer the diet a particular taxon has evolved to consume [5, 8, 10, 36]. When used to assign diet in this way, OPC(R) is calculated for both a taxon of unknown diet, typically a fossil, and for a comparative sample of taxa with known diets. A diet is then assigned for the unknown based on where its OPC(R) value falls in comparison to the comparative sample. Some studies of this nature, rather than measuring their own comparative sample, compare their fossil OPC(R) values to published values from previous studies [19, 22, 24, 37]. Both cross-study and within-study comparison of OPC(R) values have varied widely in the comparative samples chosen, in some cases using close relatives to the unknown [24, 37] and in other cases choosing a sample for assumed dietary analogy [19, 22]. No standard for comparison has been adopted here as it has been for other fields that use techniques such as the extant phylogenetic bracket [38]. Intrinsic to many of these studies, then, is some level of assumption that the dietary signal seen in OPCR is stronger than the effects of taxonomic or methodological variation.

Yet, the relative impact of dietary signal compared to these other factors has not been established. The 3D tooth models used for OPC(R) need to be cropped, smoothed, and standardized in polygon count, and variation in any of these methods has been found to affect results [39]. No consensus has ever been reached on how to carry out these necessary processes, and in fact many studies select entirely new combinations of methods that should make direct comparisons of results impossible (Table 1). The way diet is classified could also have an effect. The most common method is to sort diets into broad categories of foods that the animals have adapted to eat, which may or may not have a quantitative basis [40]. The classifications used in the initial Evans et al. [3] paper included "herbivore", "carnivore", and several omnivorous diet categories in between; however, this and similar types of classification have been found to

**Table 1. Methodological variation of studies included in meta-analysis.**

| Source | Teeth studied | Upper/Lower | OPC/OPCR | Cropping method | Data resolution |
|---|---|---|---|---|---|
| Evans et al. 2007 [3] | Carnassials and molars | Both | OPC | 2.5D | 150 data rows |
| Bunn et al. 2011 [2] | Molar 2 | Lower | OPCR | EEC | 10000 polygons |
| Santana et al. 2011 [21] | All teeth | Both | OPC | 2.5D | 150 data rows |
| Tiphaine et al. 2013 [36] | Molar 1 | Upper | OPC | EEC | 575 data rows |
| Melstrom 2016 [9] | All teeth | Lower | OPCR | Crown above gumline | 25 data rows/tooth |
| Ungar et al. 2016 [45] | Molar 2 | Upper | OPCR | BCO | 50 data rows/tooth |
| Pineda-Munoz et al. 2017 [5] | Postcanines | Both | OPCR | 2.5D | 50 data rows/tooth |
| Smith 2017 [52] | Penultimate molar | Lower | OPCR | EEC | 10000 polygons |
| Spradley 2017 [10] | Molar 2 | Lower | OPCR | EEC | 10000 polygons |
| Berthaume et al. 2019 [39] | Molar 2 | Lower | OPCR | EEC | 10000 polygons |
| Fulwood 2019 [35] | Molar 2 | Lower | OPCR | EEC | 10000 polygons |
| Pérez-Ramos et al. 2020 [50] | Premolar 4 and molars | Upper | OPCR | EEC | 10000 polygons |
| Selig et al. 2020 [51] | Molars | Lower | OPCR | EEC | 10000 polygons/tooth |
| Christensen and Melstrom 2021 [46] | All teeth | Both | OPCR | Crown above gumline | 25 data rows/tooth |
| López-Aguirre et al. 2021 [8] | Molar 1 | Lower | OPCR | BCO | Unspecified |
| Selig et al. 2021 [29] | Molar 2 | Lower | OPCR | EEC | 10000 polygons |
| Waldman et al. 2023 [53] | Postcanines | Lower | OPCR | EEC | 10000 polygons |

Highlighted cells indicate shared methods. Studies sorted by publication year. "2.5D" indicates studies in which teeth were not cropped, but were instead studied from a top-down perspective with height data instead of from a 3D mesh. "EEC" indicates that the entire enamel cap was analyzed. "BCO indicates that the tooth mesh was cut off at the point of the lowest tooth basin.

obscure much of the dietary variation in mammals [40]. In particular, the conflation of different types of faunivory is a point of concern, because invertivores consume food with far different material properties than vertebrate flesh, a difference reflected in the tooth morphology [41–44]. Finally, OPC(R) results might also not be comparable due to taxonomic differences. Although the pattern of higher values in herbivores and lower values in carnivores has been observed in multiple amniote groups, the range of values taken and scale of difference among categories studied can vary enormously across taxa. For example, a single bat tooth can have more than 100 patches [8, 21], a primate tooth might have only around 30 [2, 39, 45], and reptile teeth can have fewer than 10 patches each [9, 46]. As such, comparing a specimen of unknown diet to an inappropriate comparative sample can produce spurious inferences.

Given the many concerns with the use of OPC(R), there is a pressing need to examine existing studies to determine where OPC(R) is effective and where it may be misleading. Here, we use meta-analysis to investigate one of the foundational ideas of the method: that herbivore teeth are more complex than faunivore teeth [3]. The meta-analytic effect size applied here is standardized mean difference (SMD) [47], which we use to standardize and compare the difference in OPC(R) between herbivores and faunivores in all published studies for which these data are available. We compare SMD among samples involving different types of herbivory and faunivory, and also calculate the average SMD for terrestrial amniotes in general. From these data, we make suggestions as to how to use OPC(R) most reliably.

## Methods

### Literature search and inclusion criteria

The intent of the literature search was to identify all studies published in English that have measured OPC or OPCR in multiple species. These species also needed to be explicitly

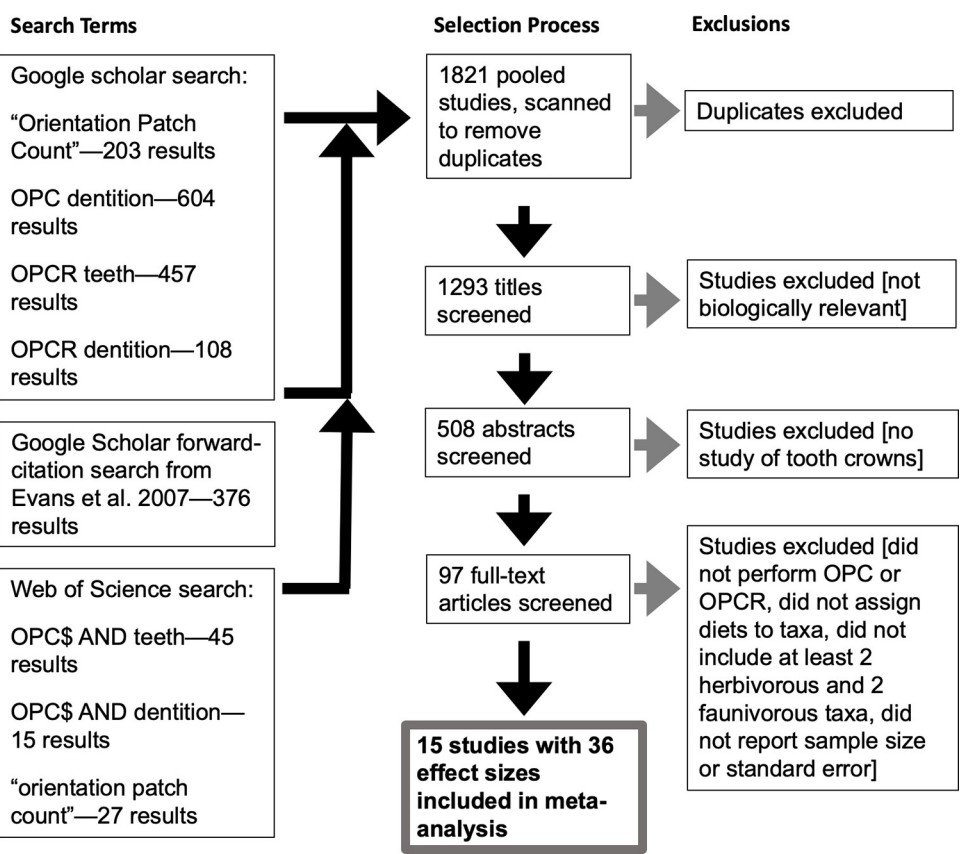

**Fig 1. PRISMA diagram showing paper screening and selection process conducted in November 2021.** Three additional studies were added to the analysis in March 2023 using a similar process, filtering for publication since 2021.

mentioned as consuming at least two different diets, at least one of which needed to be herbivorous and one of which needed to be faunivorous.

Studies were identified from the published literature using three search methods in November 2021 (Fig 1). In Google Scholar, search terms were "'orientation patch count'", "OPC dentition", "OPCR teeth", and "OPCR dentition", which returned a collective 1,372 results. In Web of Science, search terms were "OPC$ AND teeth", "OPC$ AND dentition", and "'orientation patch count'", which returned a collective 87 results. Finally, a forward-citation search from Evans et al. [3] in Google Scholar returned 376 results. Results from the three searches were pooled and duplicates were removed. This process resulted in a final count of 1293 search results.

Publication titles were screened to exclude non-English-language content and studies in disciplines unrelated to biology, reducing the results to 508 studies. The abstracts of these studies were then screened for relevance to dental morphology. At the same time, dates were scanned for publication in 2007 or later, reducing the results to 97 studies. Finally, these 97 studies were individually screened for OPC(R) results calculated for multiple species that were explicitly described as having distinct diets including some type of herbivore and some type of faunivore. We found 16 studies satisfying these requirements (Table 1); however, one did not report variances for OPC(R) measurements and could not be included in the meta-analysis [48]. One additional paper was excluded because it was redundant in sample with another study by the same authors [49]. Searches were repeated in March 2023, filtering for studies published since November 2021. Three studies were added using this method (Table 2).

**Table 2. Studies included in analysis.**

| Source | Sample number | Study group | Herbivore diet | Faunivore diet | SMD |
|---|---|---|---|---|---|
| Berthaume et al. 2019 [39] | 1 | Prosimians | Folivore | Insectivore | -0.3 |
| Berthaume et al. 2019 [39] | 2 | Prosimians | Frugivore | Insectivore | -0.1 |
| Bunn et al. 2011 [2] | 1 | Primates | Folivore | Insectivore | -1.3 |
| Bunn et al. 2011 [2] | 2 | Euarchontans | Frugivore | Insectivore | -7.5 |
| Christensen and Melstrom 2021 [46] | 1 | Squamates | Herbivore | Carnivore | 1.77 |
| Christensen and Melstrom 2021 [46] | 2 | Squamates | Herbivore | Insectivore | 1.05 |
| Evans et al. 2007 [3] | 1 | Carnivorans | Herbivore | Carnivore | 3.21 |
| Evans et al. 2007 [3] | 2 | Muroid rodents | Herbivore | Carnivore | 2.71 |
| Fulwood 2019 [35] | 1 | Strepsirrhine primates | Folivore | Insectivore | 0.45 |
| Fulwood 2019 [35] | 2 | Strepsirrhine primates | Frugivore | Insectivore | -0.3 |
| López-Aguirre et al. 2021 [8] | 1 | Noctilionoid bats | Frugivore | Carnivore-Piscivore | 0.56 |
| López-Aguirre et al. 2021 [8] | 2 | Noctilionoid bats | Frugivore | Insectivore | 0.05 |
| Melstrom 2016 [9] | 1 | Dentigerous saurians | Herbivore | Carnivore | 1.71 |
| Melstrom 2016 [9] | 2 | Dentigerous saurians | Herbivore | Insectivore | 0.91 |
| Pérez-Ramos et al. 2020 [50] | | Bears | Folivore-frugivore | Faunivore | 0.48 |
| Pineda-Munoz et al. 2017 [5] | 1 | Terrestrial mammals | Frugivore | Carnivore | 3.47 |
| Pineda-Munoz et al. 2017 [5] | 2 | Terrestrial mammals | Frugivore | Insectivore | 1.03 |
| Pineda-Munoz et al. 2017 [5] | 3 | Terrestrial mammals | Granivore | Carnivore | 2.81 |
| Pineda-Munoz et al. 2017 [5] | 4 | Terrestrial mammals | Granivore | Insectivore | 0.64 |
| Pineda-Munoz et al. 2017 [5] | 5 | Terrestrial mammals | Herbivore | Carnivore | 3.07 |
| Pineda-Munoz et al. 2017 [5] | 6 | Terrestrial mammals | Herbivore | Insectivore | 0.87 |
| Santana et al. 2011 [21] | | Phyllostomid bats | Frugivore | Insectivore | 2.72 |
| Selig et al. 2021 [29] | 1 | Euarchontans | Folivore | Insectivore | 1.64 |
| Selig et al. 2021 [29] | 2 | Euarchontans | Frugivore | Insectivore | -0.8 |
| Selig et al. 2020 [51] | 1 | Euarchontans | Folivore | Insectivore | 1.92 |
| Selig et al. 2020 [51] | 2 | Euarchontans | Frugivore | Insectivore | 0.29 |
| Smith 2017 [52] | 1 | Terrestrial mammals | Folivore | Insectivore | -2.8 |
| Smith 2017 [52] | 2 | Terrestrial mammals | Folivore | Carnivore | -0.4 |
| Smith 2017 [52] | 3 | Terrestrial mammals | Frugivore | Insectivore | 7.02 |
| Smith 2017 [52] | 4 | Terrestrial mammals | Frugivore | Carnivore | 17.1 |
| Spradley 2017 [10] | 1 | Marsupials | Folivore | Faunivore | 1.88 |
| Spradley 2017 [10] | 2 | Marsupials | Folivore | Insectivore | 1.28 |
| Spradley 2017 [10] | 3 | Marsupials | Frugivore | Insectivore | -0.5 |
| Spradley 2017 [10] | 4 | Marsupials | Frugivore | Faunivore | 0.8 |
| Tiphaine et al. 2013 [36] | 1 | Muroid rodents | Frugivore-granivore | Piscivore | 1.44 |
| Tiphaine et al. 2013 [36] | 2 | Muroid rodents | Frugivore-granivore | Insectivore | 2.11 |
| Tiphaine et al. 2013 [36] | 3 | Muroid rodents | Folivore | Piscivore | 3.74 |
| Tiphaine et al. 2013 [36] | 4 | Muroid rodents | Folivore | Insectivore | 7.55 |
| Ungar et al. 2016 [45] | 1 | Platyrrhine primates | Folivore-frugivore | Insectivore-gumivore | 1.31 |
| Ungar et al. 2016 [45] | 2 | Platyrrhine primates | Frugivore-granivore | Insectivore-gumivore | 1.27 |
| Ungar et al. 2016 [45] | 3 | Platyrrhine primates | Frugivore-folivore | Insectivore-gumivore | 2.73 |
| Ungar et al. 2016 [45] | 4 | Platyrrhine primates | Hard-object frugivore | Insectivore-gumivore | 1.77 |
| Waldman et al. 2023 [53] | | Carnivorans | Herbivore | Carnivore | -1 |

Sample number distinguishes among samples taken from the same source. Herbivore and faunivore diets are labelled according to usage in the source publication. SMD is the standardized mean difference between herbivore and faunivore mean OPC(R).

**Table 3. Diet categories seen in samples.**

| Diet categories | Faunivore/Herbivore category | Primary food items |
|---|---|---|
| Folivore | Herbivore | Leaves and stems |
| Folivore-frugivore | Herbivore | Leaves, stems, and fruit |
| Frugivore | Herbivore | Fruit |
| Frugivore-granivore | Herbivore | Fruit and seeds |
| Granivore | Herbivore | Seeds |
| Hard-object frugivore | Herbivore | Resistant nuts and fruits |
| Herbivore | Herbivore | Any plant material |
| Carnivore | Faunivore | Varies according to usage in publication, either any animal tissue or specifically vertebrate flesh |
| Carnivore-Piscivore | Faunivore | Flesh of both terrestrial vertebrates and fishes |
| Insectivore | Faunivore | Arthropods, both hard-shelled (e.g. beetles) and pliant (e.g. moths) |
| Insectivore-gumivore | Faunivore | Insects, saps and gums of trees. Note that primary gum-consumers that secondarily consume a large proportion of insects were included in this category under the assumption that the majority of breakdown enacted by teeth is on their secondary insect food source. |

The consumed foods given here are broadly applicable, but slight variation in usage may exist among sources. The format of diet category names was standardized for presentation.

## Data extraction

For each study included in the meta-analysis, we collected data on each herbivorous and faunivorous sample group. Sample groups were defined as all specimens sharing an assigned diet within a study, and assigned diets were classified as faunivorous or herbivorous (Table 3). Data collected included the mean OPC(R) values and a measure of variance, either standard deviation or standard error. If studies included separate values for upper and lower teeth, both were collected and meta-analyses were conducted both ways, but this was found to make no difference in our results (S1 Table). Here we report only those values obtained using the lower toothrows. If a study included multiple methods of OPC(R) calculation, we used the methods that involved cropping the tooth surface to the entire enamel cap and standardizing the polygon count to 10,000, to match the most commonly used standards [39]. If multiple herbivorous or faunivorous diet groups were included in a study (for example, if the study had separate values for frugivores and folivores), these were considered as separate sample groups in the analysis. This does mean that, for some of the studies, one herbivorous sample was compared to multiple faunivorous samples, or vice versa. This similarity in sample was accounted for in the overall effect size calculation using our similarity correction, below.

From our mean OPC(R) values and estimates of variance, we calculated the standardized mean difference (SMD), also known as Hedges' *d* [47], using the R package *metafor* [54]. SMD was calculated between the mean OPC(R) of an herbivorous sample and the mean OPC(R) of a faunivorous sample within the same study, mathematically adjusted to be placed on a common scale for comparison.

Additional information was collected to be used as covariates in the meta-analysis. The specific diets for each herbivorous sample were categorized as being folivorous or not, to reflect how folivory is generally considered the most specialized and morphologically extreme form of herbivory [15]. Folivory, as defined here, includes both the consumption of browse, which

can include tender young leaves, and grass, which is tough, contains phytoliths, and is often consumed along with grit, all of which put unique demands upon the dentition [55–58]. The diets of faunivorous samples were categorized as being invertivorous or not, to reflect the distinct structural properties of invertebrates compared to vertebrate flesh [43]. It must be noted, however, that these two categorizations of diet are imperfect, because some studies do not categorize their herbivore and faunivore diets at this finer scale. Studies were distinguished as conducted on either a single tooth (usually a molar) or a larger tooth sample, and this distinction is marked in figures. Additional data collected included the type of variance, whether OPC or OPCR was used, cropping technique, and mesh polygon number or density. These variables were considered in the comprehensive review, but were not incorporated into the meta-analysis due to their association with study similarity, our low sample size, and the high likelihood of overfitting [59].

## Sample similarity correction

Typically, similarity among samples in a meta-analysis is accounted for using a mixed-effects model that includes fixed grouping effects [54], which correct for clustering in related samples. However, not all similarity in samples can be explained by clustering, even hierarchical nested clustering. One such case is where some or all the similarity is due to shared phylogenetic history. The samples in our meta-analysis are each composed of a collection of species that may overlap with species in other samples, and all of which vary in phylogenetic relatedness. To incorporate this type of structured non-independence, we adapted a measure of similarity typically used to describe community similarity in ecological studies, the mean nearest taxon distance [60, 61].

The mean nearest taxon distance is computed by finding the minimum length of branches on a phylogeny separating a taxon in one group from any of the taxa in the other group, and then averaging those distances for all taxa in both groups [61]. Usually, the groups in question are ecological communities, but here our groups are sample groups, made up of one herbivorous and one faunivorous sample from a study. Mean nearest taxon distance has the advantage over other measures of group similarity in that it directly accounts for species overlap, with overlapping species having a distance of 0. Because some species in our analysis were examined in more than one study, accounting for this overlap was necessary.

We created a phylogeny of all species in our sample groups using TimeTree (S1–S6 Figs) [62]. The phylogeny was used to calculate mean nearest taxon distances for each pair of sample groups in our analysis (Fig 2), which were then organized into a community distance matrix using the R package *picante* [63]. The community distance matrix was then converted into a correlation matrix. It has been previously demonstrated by Adams [64] that meta-analytic data can be transformed using a correlation matrix to account for structured similarity, which has previously been used to account for phylogenetic similarity. We adapt the methods of Adams to transform our SMD according to the correlation of mean nearest taxon distance among our studies.

## Assessment of publication bias

With the exception of a few studies where demonstrating differences in OPC(R) between herbivores and faunivores was among the main published takeaways of the work [3, 9, 21, 46], most of the studies in our analysis included association between OPC(R) and diet as only part of a larger suite of analyses. As such, they are unlikely to be subject to the form of publication bias in which the entire study is suppressed because of non-significant results [65]. However, our sample of studies remains susceptible to bias in that studies may be less likely to include

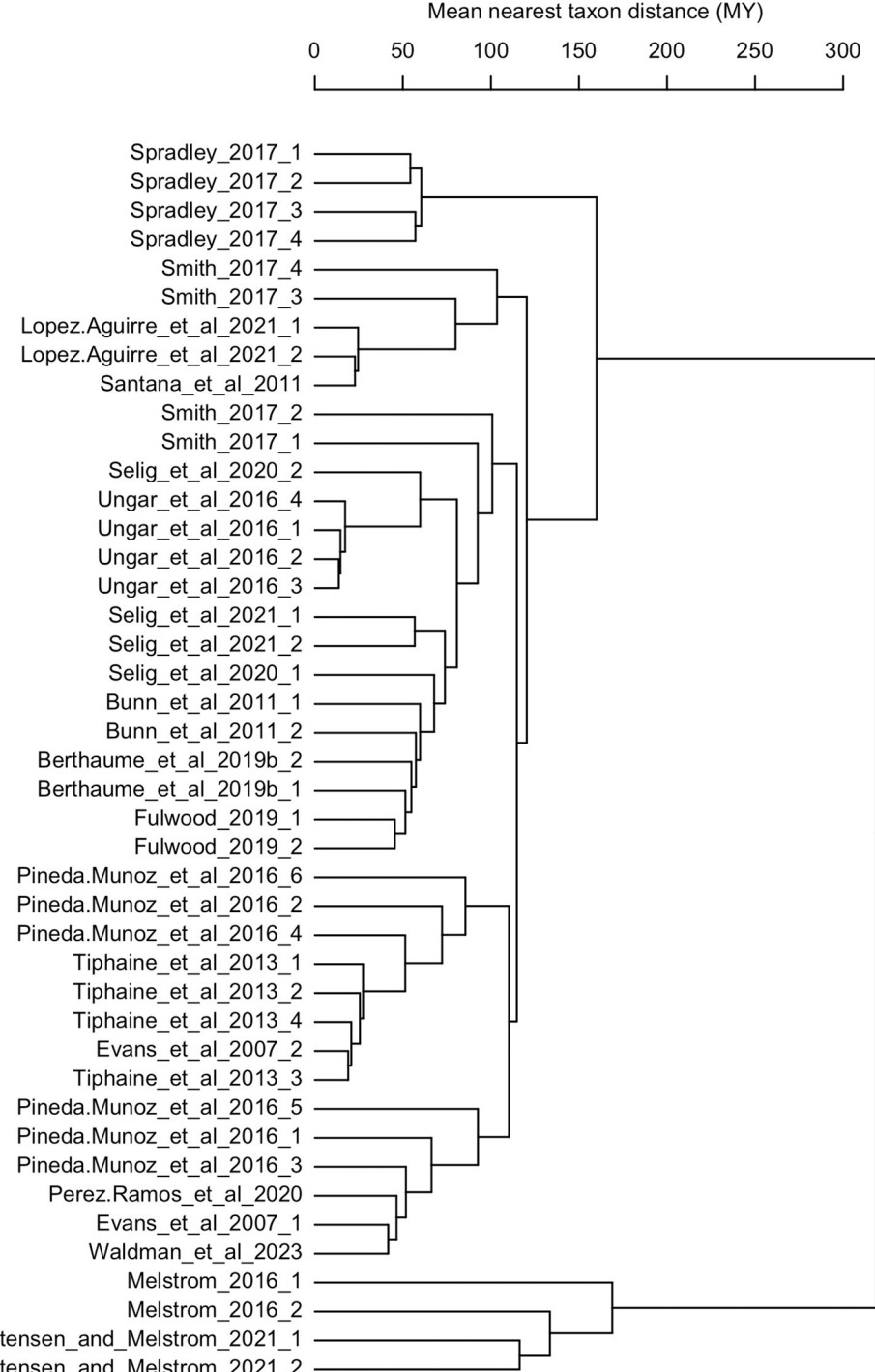

**Fig 2. Dendrogram showing similarity among sample groups included in meta-analysis, measured in mean nearest taxon distance.** Scale is in units of millions of years, indicating the average number of years of divergence for the taxa in one sample group to the mostly closely-related taxa in other sample groups.

OPC(R) values if such values were found to not be useful in distinguishing among diets. For this reason, we examined our sample of studies to infer whether publication bias may be skewing the effect sizes in our meta-analysis. We created funnel plots of our data both before and

after phylogenetic correction [66] using the *metafor* package [54], to help visualize the distribution of effect sizes as compared to study precision. We also arranged forest plots of effect sizes in order of precision [67] to further check for systematic differences in effect size between larger, more statistically robust studies and smaller studies. These methods can help visually identify whether there is suppression of effects with negative or non-significant results.

### Phylogenetic signal

To test the strength of phylogenetic signal in OPC(R), we examined the dataset of Pineda-Munoz et al. [5], one of the larger datasets of mammals (*n* = 106) used in our study (omitting *Ichthyomys hydrobates*, *I. stolzmanni*, *Notamacropus dorsalis*, *and N. rufogriseus*, for which genetic data were not available). We calculated Blomberg et al.'s [68] *K* using the *phytools* package [69] and a phylogeny created using TimeTree [62]. *K* is a measure of the degree to which variation in the measured outcome corresponds to the structure of the phylogenetic tree, with *K* = 1 indicating variation consistent with Brownian motion along the tree and *K* = 0 indicating random variation with respect to the tree. We tested whether our calculated *K* values significantly differed from 1 and 0 based on 10000 simulations of Brownian motion along the branches of the tree using the *phytools* package [69]. This was done for both average and total OPCR of the tooth row. We also tested the phylogenetic signal for only the rodents (*n* = 49) in this data set, for an example of how phylogenetic signal might change when the breadth of the group of interest is reduced.

### Meta-analysis

We performed a meta-analysis using our calculated SMD values and associated variances. Because OPC(R) was initially proposed as following a common pattern across mammals [3], and has been put forward as following the same pattern in other amniotes, we first fitted a fixed-effect model to our untransformed SMD. We did this using a generalized linear model in R [70] weighted by the inverse variances of our data.

To more realistically model our data, we fitted two additional models. The first was a fixed-effects model of SMD values that had been transformed using our measure of study similarity (above), to demonstrate the effect of accounting for phylogenetic similarity and study overlap. The second was a mixed-effects model, also performed on the transformed SMD values, that included factors for whether the herbivore sample group was specifically folivorous or not, and whether the faunivore sample group was specifically invertivorous or not.

## Results

The usage of OPC(R) to study dietary ecology has been unevenly adopted by taxonomic group. Of the 19 studies that reported all necessary diet information, seven were analyses of primates and their close relatives [2, 29, 35, 39, 45, 49, 51], three were entirely chiropterans [8, 21, 48], and three focused on carnivorans [3, 50, 53]. Only two studies were conducted on reptiles instead of mammals [9, 46]. The remaining studies focused on muroid rodents [3, 36] and marsupials [10], and in addition there were two studies that sampled broadly across terrestrial Mammalia [40, 52].

From our funnel plot of the effect sizes of published studies (Fig 3A), we see that the distribution of data from all included studies is largely symmetrical. It generally fits a funnel shape, with the majority of values clustering together and the cluster widening as larger effect sizes are associated with larger standard errors, including one very large effect size associated with the largest standard error. It does not center at 0, instead appearing to center at a positive point. Most of this asymmetry is due to multi-tooth studies, which cluster closely at positive

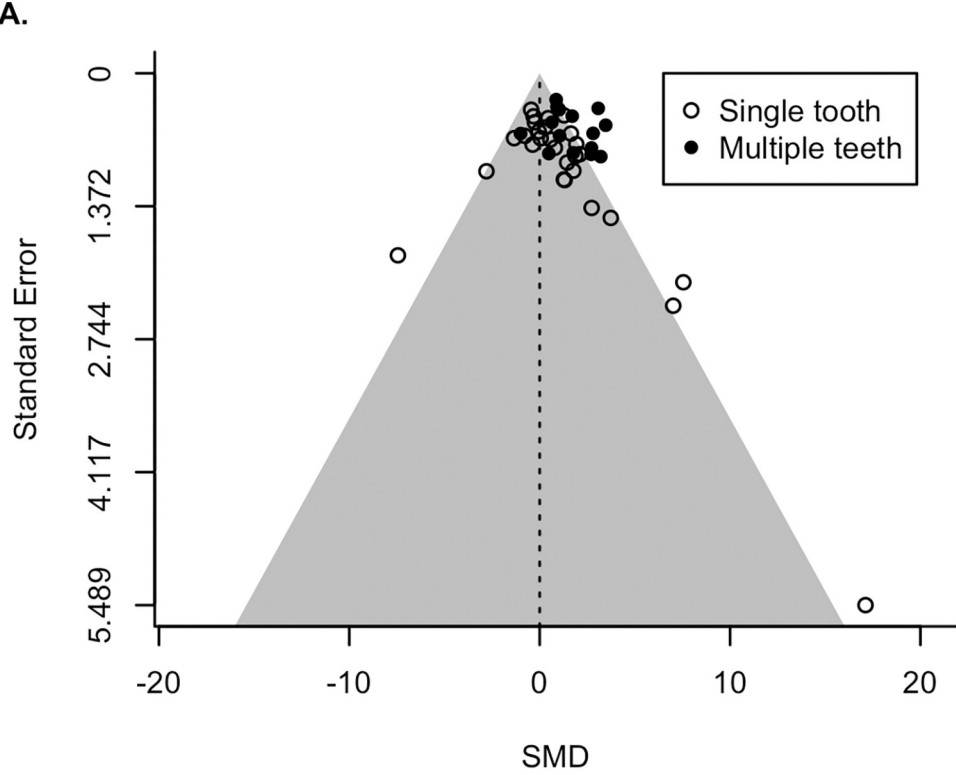

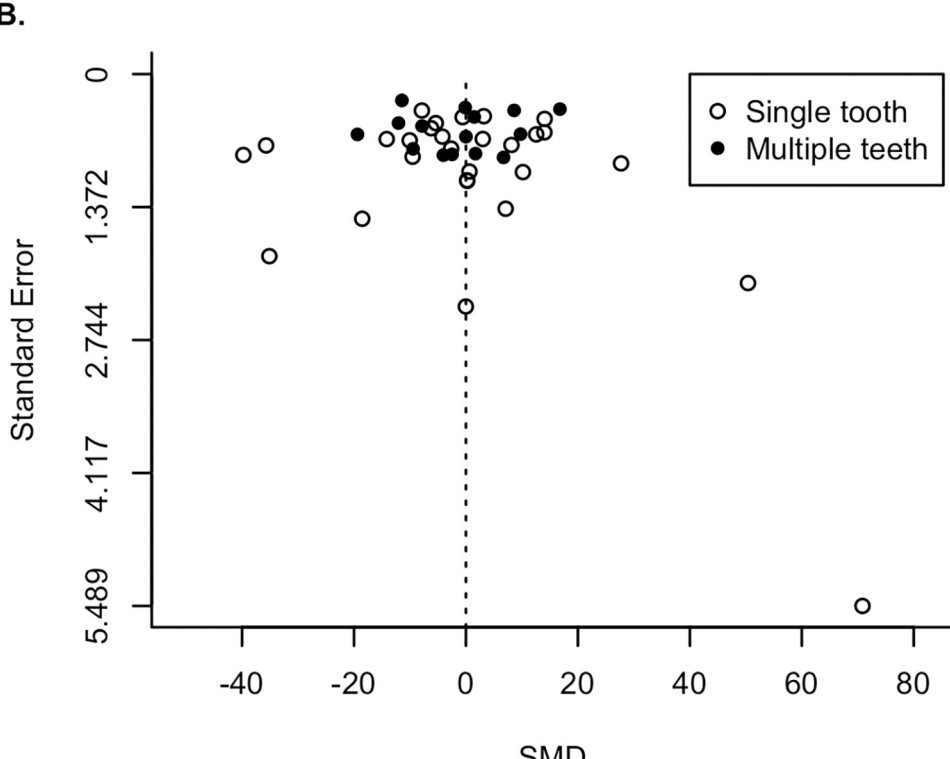

**Fig 3. Funnel plots showing the distribution of SMD compared to standard error.** SMD is the standardized mean difference. Open circles indicate studies conducted on a single tooth. Filled circles indicate studies conducted on multiple teeth. Note that standard error is shown with 0 at the top of the y axis. (A) Raw SMD values before sample

similarity correction. Shaded region represents a 99.5% pseudo-confidence interval based on standard error. (B) Adjusted SMD values after sample similarity correction.

SMD. Among single-tooth studies, there are slightly more with notably large SMDs than notably small SMDs, but these studies are all those with large standard errors. Although studies with large standard errors receive little weight in the meta-analysis, and as such should have limited ability to skew our results, we note that any finding of positive overall SMD may be influenced by publication bias against publishing OPC(R) values in small studies where they do not show the expected relationship with diet.

When studies are corrected for sample similarity (Fig 3B), the minor asymmetry seen in the funnel plot disappears. Instead, the data are now centered much closer to zero and the distribution of points appears much more randomly distributed around zero. Most notably, multi-tooth studies now cluster around zero. A similar pattern is shown in our forest plots. The forest plot of uncorrected data (Fig 4) shows that as precision increases, SMD appears to converge on

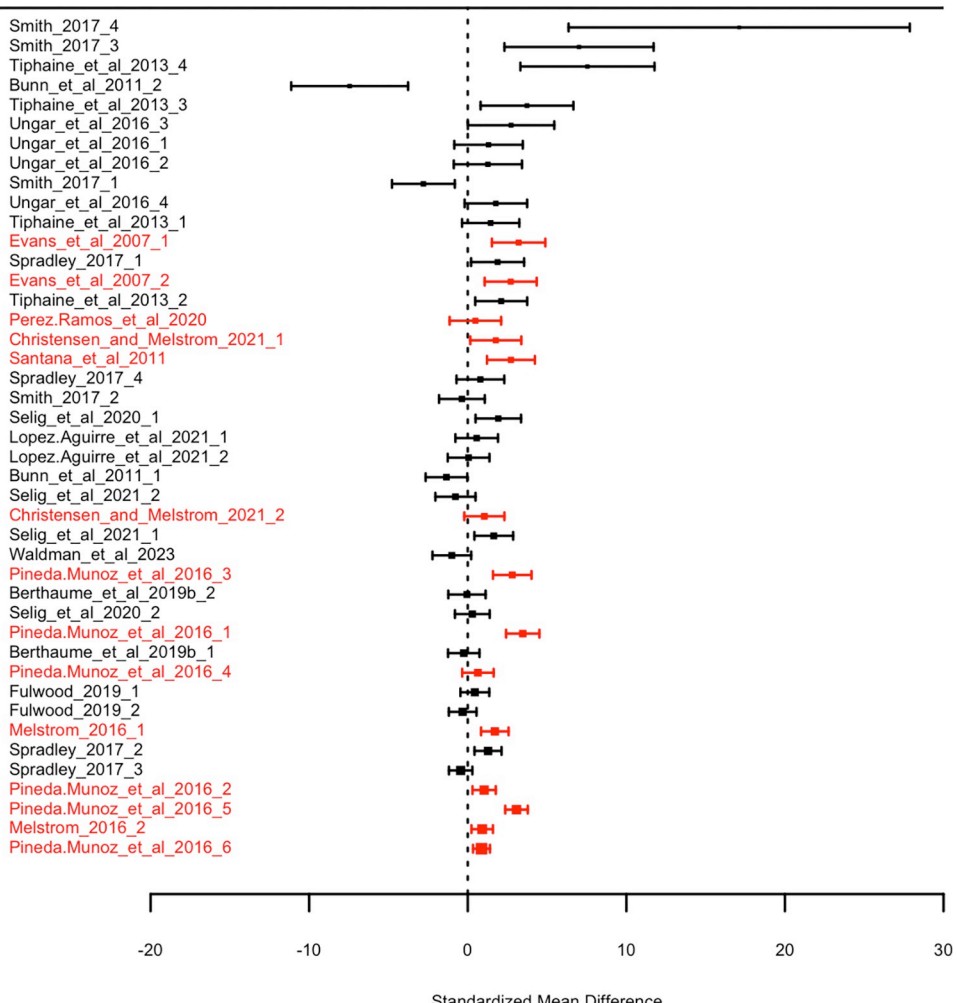

**Fig 4. Forest diagrams showing effect sizes for each study sample used in meta-analyses, before sample similarity correction.** Standardized mean difference is calculated between the mean herbivore and mean faunivore OPC(R) values. Studies of multiple teeth are shown in red, and studies of single teeth are shown in black. Size of dot indicates relative study weight. Error bars show 95% confidence intervals. Samples are ordered by increasing precision.

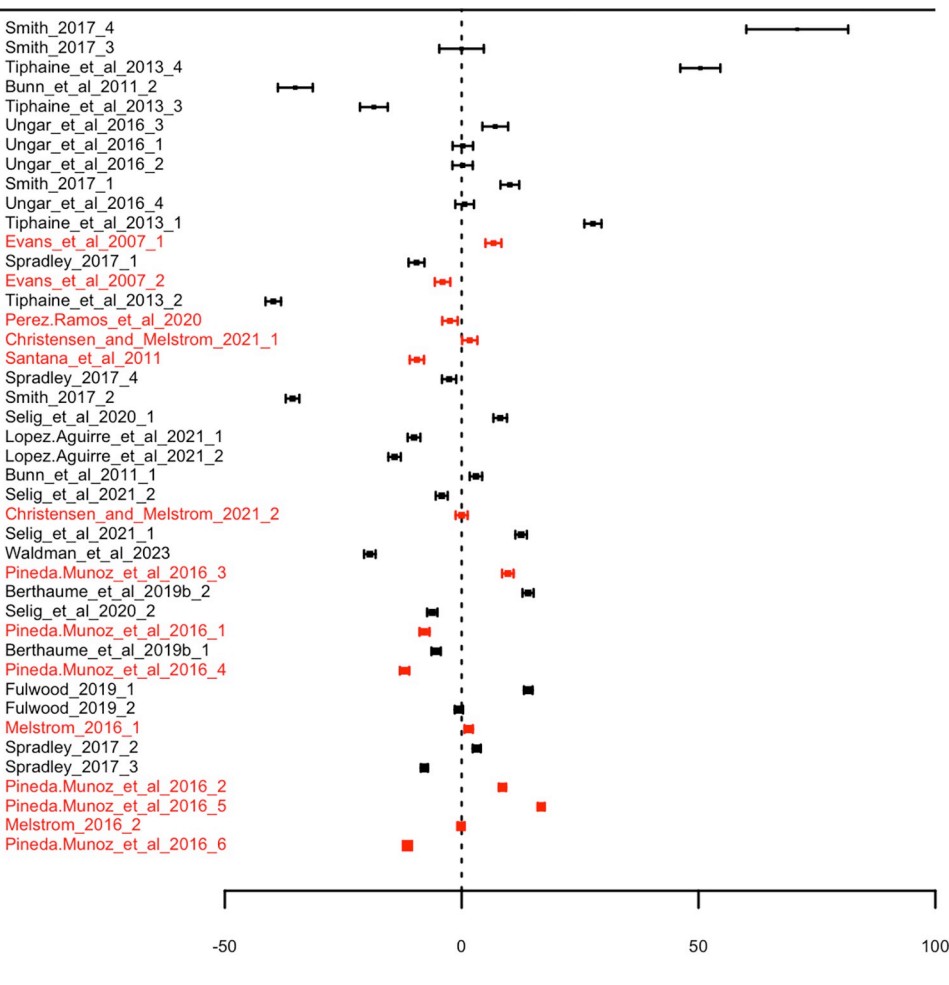

**Fig 5. Forest diagrams showing effect sizes for each study sample used in meta-analyses, after sample similarity correction.** Standardized mean difference is calculated between the mean herbivore and mean faunivore OPC(R) values, and subsequently adjusted using the mean nearest taxon distance among samples. Studies of multiple teeth are shown in red, and studies of single teeth are shown in black. Size of dot indicates relative study weight. Error bars show 95% confidence intervals. Samples are ordered by increasing precision.

a small but positive point. After sample similarity correction, there is no apparent convergence (Fig 5). This gives us confidence that results of our corrected meta-analysis should be less affected by publication bias than our uncorrected results.

The mean OPC(R) values for herbivores and faunivores vary widely across studies (Fig 6). Much of this variation is due to differences in methodology; studies looking at the total OPC (R) for multiple teeth, for example, generally have larger means than those looking at a single tooth—the overall herbivore mean OPC(R) from studies of one tooth is only 97.9, whereas the overall herbivore mean OPC(R) for studies of multiple teeth is 156.2. Because the values as presented in Fig 6 are from studies that vary in which teeth were examined, little should be inferred from these raw values, but some patterns in the ratio of faunivore mean OPC(R) to herbivore mean OPC(R) are notable. As indicated by the dotted 1:1 line, most studies support the initial relationship put forward in Evans et al. [3], which is that herbivore dentitions are generally more complex than faunivore dentitions. However, this is not universal. In ten of our samples, the faunivores have higher OPC(R), and thus more complex teeth, than the

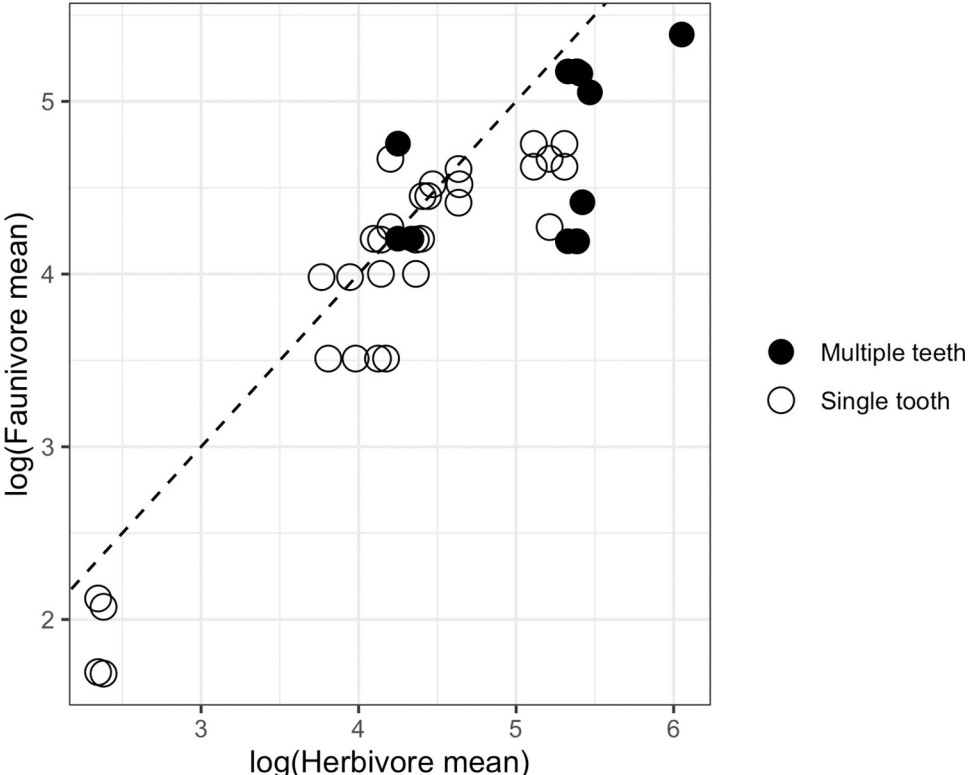

**Fig 6. Log-transformed mean OPC(R) values for herbivores and faunivores.** Each point represents a comparison of an herbivore sample to a faunivore sample within a single study, with open and filled indicating whether OPC(R) values were calculated for a single tooth or a tooth row. Dotted line shows 1:1 ratio, where points would lie if herbivores and faunivores had equal OPC(R).

herbivores. Six of these ten "reversals" are from samples that were entirely or almost entirely composed of primates [2, 29, 35, 39]. Not all samples of primates show the teeth of faunivores to be more complex than those of herbivores [29, 35, 39, 49], but primate samples in general do tend toward low mean herbivore OPC(R), compared to non-primates. The overall mean OPC(R) for single-tooth studies of primates is 68.3, and for non-primates is 123.2. Overall mean faunivore OPC(R) in studies of single teeth is also low in primates compared to non-primates, although not as dramatically (60.9 vs 86.9). The studies of primates are also notable in that all studied primate faunivores are specifically invertivores or invertivore-gummivores. Invertivores behave noticeably differently from other faunivores in the six studies where both invertivore and non-invertivore faunivore samples were available, with invertivore teeth generally being more complex. Apart from one study of muroid rodents [36] and one study of a variety of terrestrial mammals [52], where invertivore OPC(R) slightly exceeds that of other faunivores, the mean OPC(R) of non-invertivore faunivores is only 37%-82% that of invertivores in the same study.

Several species were included in multiple studies, making it possible to compare calculated OPC(R) values directly (Table 4). Four studies that included primates employed the same methods (Table 1), making direct comparison across studies feasible for 13 species [2, 29, 35, 39]. The differences in calculated OPCR for these species among studies are enormous, with some measurements being more than twice as large as others. The scale of these inter-study differences in many cases exceeds the differences between herbivores and faunivores in the same studies. It is similar, however, to the levels of variation previously observed for members

**Table 4. OPCR values for species measured in multiple studies with the same methods.**

| | Berthaume et al. 2019 [39] | Bunn et al. 2011 [2] | Fulwood 2019 [35] | Selig et al. 2021 [29] | Percentage difference |
|---|---|---|---|---|---|
| *Arctocebus calabarensis* | 85.9 | 46.8 | 101.7 | 60.5 | 74% |
| *Avahi laniger* | 95.2 | 55.6 | 105.4 | NA | 62% |
| *Cynocephalus volans* | NA | 51.5 | NA | 76.9 | 40% |
| *Euoticus elegantulus* | NA | NA | 93.9 | 57.5 | 48% |
| *Galagoides demidoff* | 78.3 | NA | 83 | 56.9 | 37% |
| *Hapalemur griseus* | 86.4 | NA | 83.7 | 80.5 | 7% |
| *Indri indri* | 69.7 | 55.6 | 100.6 | NA | 58% |
| *Lepilemur mustelinus* | 73 | NA | 85.5 | NA | 16% |
| *Loris tardigradus* | 96.4 | 52 | 95.3 | 0 | 60% |
| *Nycticebus bengalensis* | NA | NA | 94.5 | 45.1 | 71% |
| *Otolemur crassicaudatus* | NA | NA | 75.2 | 61.1 | 21% |
| *Perodicticus potto* | 83.8 | 51.8 | 85.2 | NA | 49% |
| *Prolemur simus* | 85.3 | NA | 150.5 | NA | 55% |

Percentage difference is calculated by comparing the largest to the smallest calculated OPC(R) for that species, where 0% would indicate perfect agreement among studies.

of the same species with differing levels of tooth wear [71]. The scale of these differences is also not consistent among studies. For example, the OPC(R) calculated for *Hapalemur griseus* by Fulwood [35] only differs by 7% from the value calculated for that species by Selig et al. [29], but the OPC(R) calculated for *Nycticebus bengalensis* by Fulwood is over twice as large of that calculated by Selig et al.

To test how phylogeny might influence OPC(R) results, we calculated Blomberg et al.'s [68] *K* for the dataset of Pineda-Munoz et al. [5]. For the total tooth row, *K* was calculated to be 0.452. This is significantly higher (p < 0.0001) than a result that would be expected if OPCR values were random with respect to the tree and not significantly lower (p = 0.20) than would be expected if OPCR values were entirely determined by Brownian motion along the branches of the phylogeny, based on 10000 simulations. For average OPCR, *K* is 0.292, which is again significantly higher (p < 0.0001) than we would expect if OPCR were random with respect to phylogeny, but marginally significantly lower (p = 0.04) than would be expected under Brownian motion on the phylogeny.

When this analysis was re-run for only rodents, *K* was found to be 0.707 for the total tooth row and 0.785 for the average tooth. These are both significantly (p = 0.0012 and p = 0.0003, respectively) higher than expected if OPCR was random with respect to the phylogeny, and not significantly lower (p = 0.3494 and p = 0.4662) than would be expected under Brownian motion on the phylogeny.

The results of our meta-analyses varied according to the data and model used. The fixed-effect model that was not adjusted for study similarity calculated an overall effect size of 0.97, with a 95% confidence interval of 0.56 to 1.38, which is significantly greater than 0. As such, the most naïve interpretation of our data that does not consider sample similarity or within-diet variation does find greater complexity in the teeth of herbivores than faunivores.

The fixed-effect model based on data adjusted for sample similarity instead finds an estimated overall effect size of -8.08, with a 95% confidence interval of -19.01 to 2.84. With a negative estimated effect size and a confidence interval that overlaps 0, we cannot conclude based on these data that the true difference between herbivore and faunivore means for amniotes is greater than 0. This result is echoed by our mixed-effects model that includes factors for herbivore and faunivore diet, which finds an estimated effect size of -13.18 and a 95% confidence

interval that also overlaps 0 (-30.95 to 1.61). This model also does not find significant effects of herbivore or faunivore diet type on effect size, with p-values of 0.7 and 0.13, respectively. These results indicate that there is a significant difference found between the pooled herbivore and pooled faunivore diet mean OPC(R) values only when the confounding effects of phylogeny and study similarity are ignored.

## Discussion

The high variation in calculated OPC(R) across studies is concerning for those interested in comparing new results to published values. Although it has been established for years that methodology is critical to replicability with OPC(R) [39], the extreme variation seen even among studies employing the same methods (Table 4) raises the question of whether cross-study comparison is ever valid. Some of the observed variation may be due to actual intraspecific differences, which can be quite large if the level of wear varies among individuals [71], suggesting that there is a need for more robustly controlling the level of wear of specimens chosen for OPC(R) analysis. Unfortunately, at this time, too few studies have attempted to repeat OPC(R) analyses on species previously studied for us to distinguish the effects of intraspecific variation from other possible unaccounted-for variations in methodology. For now, we recommend making any cross-study comparisons with extreme caution. Where methods are not identical, we do not recommend cross-study comparison be performed at all. Where methods are the same, raw or mean values should still not be assumed to be comparable across studies without prior validation. If cross-study comparison is desired, we recommend performing such validation by calculating OPC(R) for one or more of the specimens examined in previous studies and noting differences in scale and variance. Such comparisons should become more accessible in the future with the increasing use of online repositories such as MorphoSource (www.morphosource.org) for sharing three-dimensional scans.

Even if cross-study comparison is not the goal, we still recommend caution in the application of OPC(R). Methods in OPC(R) analysis have never been standardized (Table 1), and as such, we do not recommend looking at OPC(R) analysis as a single technique producing a single inference, but instead as a collection of related techniques that must be carefully considered before being applied to any problem. Measuring multiple teeth, for example, appears to capture a different signal than measuring any single tooth, and may in fact be better suited to detecting differences between faunivores and herbivores than measuring a single tooth, although this has been tested relatively few times (Fig 3A). Because of the strong phylogenetic signal found in the dataset of Pineda-Munoz et al. [5], we also recommend that the phylogenetic scope of any OPC(R) analysis be chosen carefully. This is especially relevant for those seeking to compare the values of modern and fossil taxa for which standard phylogenetic analyses may be more difficult. At present, we should be cautious in assuming that the range of OPC(R) values taken by any fossil taxa would correspond to the same dietary signal seen in living amniotes.

Although it could not be tested directly here, due to the low number of studies on any individual taxon, we find it highly likely that patterns in OPC(R) are taxon-specific. Taxon-specific patterns are already known for other tooth metrics, such as mesowear and shearing ratio [72–74], and as such it is known that the same dietary relationships can be expressed differently in different taxa. The most obvious indicator that this is in play here is in the extremely low mean OPC(R) values for the reptiles in our study compared to the mammals. Reptile teeth are simpler than mammal teeth for any given diet. If the teeth of reptilian herbivores are more complex than those of reptilian faunivores, as has been supported in the two studies on this taxon so far [9, 46], this pattern would occur over a range of values not applicable to studies of mammals. Similarly, such patterns might exist within mammalian sub-groups. Muroid rodents, for

example, have been the focus of two studies, both of which have found that every sample group of herbivores has higher OPC(R) than the sample groups of faunivores, including invertivores [3, 36]. Although the ranges of OPC(R) values corresponding to diet in these two studies do not match due to methodological differences, it is feasible that there is a true association of certain levels of tooth complexity in this clade to herbivory and faunivory. The clade where we feel least confident in connecting OPC(R) to herbivory and faunivory is the primates. Of the fourteen sample groups tested here for primates [2, 29, 35, 39, 45, 51], six show faunivore mean OPC(R) as greater than herbivore mean OPC(R). This is substantial reversal of the pattern initially indicated by Evans et al. [3], and indeed, these studies do not attempt to assert that such a pattern is present. If OPC(R) is connected to diet in primates, that connection does not likely take the form of higher values in herbivores than faunivores, at least not for the teeth that have been tested thus far.

Part of what leads to the low contrast between herbivore and faunivore values in primates is that all primate faunivores studied are invertivores. Across studies of both primates and other mammals, the OPC(R) of invertivores was almost universally higher than other faunivores, often being double or more the OPC(R) of other faunivores. This higher complexity seen in invertivore teeth compared to other faunivores is unsurprising. Bunn et al. [2], in a study of primate second molars, found no significant difference in OPCR between insectivores and folivores. The similarity of invertivore and folivore teeth has been noted for decades, with Kay [41] being first to quantify similarities among functional features of the molars of insectivorous and folivorous primates, finding that body size was the only reliable way to discriminate between them. This similarity has been explained by referencing the similar material properties of leaves and insects, in that both resist crack propagation in ways not seen in other foods [43] and tend to require specialized shearing crests for breakdown [42, 75]. The similarity between invertivore and herbivore teeth may not hold for all taxa, however. The Tiphaine [36] study of muroid rodent upper molars found that the invertivores examined had more complex teeth than other faunivores. Currently, it is not clear whether this difference is due to the highly distinct dentitions of rodents [76], to body sizes in relation to insect prey, or to some other factor. Most of the literature on the dental similarity of invertivores to herbivores has been conducted on primates, leaving the relationship in other taxa unknown.

Our analyses did not recover a significant difference in OPC(R) between herbivores and faunivores for terrestrial amniotes in general, except when the effects of similarity among samples are ignored. It has previously been noted that accounting for similarity due to phylogeny can reduce the accuracy of dietary inference [77], which is likely due to many extant amniotic clades being ecologically specialized [76, 78, 79]. With the low breadth of taxa that have been the subject of OPC(R) analysis, and the often-small differences among diets, we do not currently have evidence to make any general statements about the existence of a relationship between OPC(R) and diet for terrestrial amniotes. Despite early evidence for the universality of OPC(R) patterns across taxon boundaries [3], subsequent publications have not supported such a generalization. Studies within some taxa, however, such as reptiles [9, 46], muroid rodents [3, 36], and bats [8, 21], have thus far shown support for a pattern of higher OPC(R) in herbivores than faunivores. In these and other taxa where such a pattern is found, distinguishing faunivores and herbivores based on OPC(R) remains feasible, as long as inferences are not extended beyond the taxon being studied.

The lack of generalizability of OPC(R)-based inference raises problems for researchers extending the technique to fossils. If patterns in OPC(R) and diet can only be confirmed within individual crown clades, then direct dietary inference for fossils should be limited to those that are nested within crown clades. For some research, this is sufficient. By comparing extinct members of extant clades to their modern relatives where patterns in OPC(R) are known,

strong inferences are possible [8, 36, 50]. For those fossil taxa that fall outside of crown clades, we recommend a more comparative method. Instead of directly assigning diets to extinct taxa, the OPC(R) of lineages can be compared over space, or time. It is then possible to make inferences about trophic structure or evolutionary change based on these comparisons, without imposing assumptions about functional analogy that may not be supported. Such a strategy has been used to good effect in the past [22–30].

The widespread adoption of OPC(R) in tooth analyses is a testament to its utility as a tool for quantifying the complexity of tooth crowns, a utility that has in the past decade been extended beyond the broad discrimination of faunivores and herbivores. Although there may be no universal pattern of OPC(R) and diet across terrestrial amniotes, patterns can still be found within individual clades. Even where we lack the evidence to support any specific relationship of OPC(R) to diet, it remains a useful tool for comparing complexity in dentitions across space and time.

## Supporting information

**S1 Checklist. PRISMA checklist.**
(PDF)

**S1 Table. Results of meta-analyses using only lower or only upper tooth rows from studies where both were available.** Asterisks (*) indicate significant difference from 0.
(PDF)

**S2 Table. Complete data used for review and analysis.**
(PDF)

**S1 Fig. Phylogenetic tree of species from all samples used in analysis, with major clades labeled.** Made using TimeTree (Kumar et al. 2022).
(PDF)

**S2 Fig. Subset of phylogenetic tree used in analyses, clade Reptilia.** Made using TimeTree (Kumar et al. 2022).
(PDF)

**S3 Fig. Subset of phylogenetic tree used in analyses, clade Euarchonta.** Made using Time-Tree (Kumar et al. 2022).
(PDF)

**S4 Fig. Subset of phylogenetic tree used in analyses, clade Rodentia.** Made using TimeTree (Kumar et al. 2022).
(PDF)

**S5 Fig. Subset of phylogenetic tree used in analyses, clades Chiroptera, Carnivora, and affiliated taxa.** Made using TimeTree (Kumar et al. 2022).
(PDF)

**S6 Fig. Subset of phylogenetic tree used in analyses, clade Marsupialia.** Made using Time-Tree (Kumar et al. 2022).
(PDF)

## Acknowledgments

We are grateful to Steve Hovick for sharing his expertise in the methods of meta-analysis, and to Jonathan Calède, Debbie Guatelli-Steinberg, and Jill Leonard-Pingel for their feedback and

suggestions regarding both the work and the manuscript. This paper was submitted by ACD in partial fulfillment of requirements for the degree Doctor of Philosophy in Evolution, Ecology, and Organismal Biology at The Ohio State University.

## Author Contributions

**Conceptualization:** Anessa C. DeMers, John P. Hunter.

**Data curation:** Anessa C. DeMers.

**Formal analysis:** Anessa C. DeMers.

**Investigation:** Anessa C. DeMers.

**Methodology:** Anessa C. DeMers.

**Supervision:** John P. Hunter.

**Writing – original draft:** Anessa C. DeMers.

**Writing – review & editing:** John P. Hunter.

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
