## [Decision Letter · Decision Letter 0]

18 Jul 2023

PONE-D-23-17989Dental complexity and diet in amniotes: a meta-analysisPLOS ONE

Dear Dr. DeMers,

Thank you for submitting your manuscript to PLOS ONE. After careful consideration, we feel that it has merit but does not fully meet PLOS ONE’s publication criteria as it currently stands. Therefore, we invite you to submit a revised version of the manuscript that addresses the points raised during the review process.

Please try to address comments in relation to the phylogenetic signal in OPC and OPC(R) datasets together with diet definitions. Try to provide a clear labelling and separation for studies looking at single vs multiple teeth and if possible explore more quantitatively the phylogenetic signal if dataset with n>20 allows.Check carefully also some figure and tables to accommodate and clarify reviewers points of concern. 

We look forward to receiving your revised manuscript.

Kind regards,

Carlo Meloro

Academic Editor

PLOS ONE

Journal Requirements:

3. We note that this manuscript is a systematic review or meta-analysis; our author guidelines therefore require that you use PRISMA guidance to help improve reporting quality of this type of study. Please upload copies of the completed PRISMA checklist as Supporting Information with a file name “PRISMA checklist”.

Additional Editor Comments:

This paper is very well executed and I really enjoyed reading it. The methodological approach is robust and the conclusions are very intriguing. What I am a bit concerned is the comparison of OPC and OPCR data by merging single tooth with multiple teeth dataset. I think it will make more sense to separate these two types of data and try labelling in a more coherent order especially in the figures of the manuscript (e.g. Fig. 3, and or Fig. 5 and 6).

Another issue I would like to be expanded is the realisation of what could be the most standard approach to collect and process OPC and OPCR data. In table 2 it is quite evident that different authors might publish different values for the same species. But still, it is not clear how much biological should be the variation observed within or between species. For instance, Cynocephalus Volans present values of 51.5 and 76.9….are these values coherent if applying OPC(R) to two different specimens of the same species? How much disparate the values are expected to be within the same study. Maybe reporting parameters such as CV within species under the same study setting could be good to present and discuss. It is pretty clear to me that merging these dataset are extremely difficult, but if someone decide to set up a new study, what kind of approach should use to make the results as much compatible as possible with other kind of work.

For Figure 2 (cluster analyses of paper) please provide a label for the X axis (is it euclidean distances, based on what?). Also colour code studies citations based on single vs multiple teeth and if eventually possible at all -based on taxonomy. We expect some sort of clustering based on these two factors.

In Figure 3 it will be good to see transverse lines delimiting the fennel. Also I presume it is standard practise that the Standard Error axis is reversed (going from positive values to zero), or not? Please explain better in the figure caption. My other concern is the quite large range of Effect Size…it might be possible that maybe the values compared are just too large…did you try by log transforming them?

Figures 5 and 6 do not seem to reflect much about what it is said in the text (e.g. line 358-360) that present values difficult to relate to the figures. Can you perhaps present an additional figure to show the fixed-effect model more efficiently?

In Table 2 the percentage difference is presented in a format that is a bit counterintuitive. The percentage difference between two studies showing exactly the same values should be 0% and not 100% in my opinion. Also, try to swap the raw with column so that the Table can be read more effectively.

Last point, I will be curious to see the phylogenetic signal (as in Blomberg, S. P., Garland Jr, T., & Ives, A. R. (2003). Testing for phylogenetic signal in comparative data: behavioral traits are more labile. Evolution, 57(4), 717-745) in the OPC and OPC(R) if dataset allows (n > 20). I think the Evans et al rodent and carnivorans data should allow that. It would be good to enforce the hypothesis that OPC and OPC(R) are not independent of phylogeny. Within mammals, it will also be interesting to see what could be the best starting dataset and the best data to merge so that for future refs. people have a clear idea on how data should vary.

Reviewers' comments:

Reviewer's Responses to Questions

**Comments to the Author**

1. Is the manuscript technically sound, and do the data support the conclusions?

Reviewer #1: Yes

2. Has the statistical analysis been performed appropriately and rigorously? 

Reviewer #1: Yes

3. Have the authors made all data underlying the findings in their manuscript fully available?

Reviewer #1: Yes

4. Is the manuscript presented in an intelligible fashion and written in standard English?

Reviewer #1: Yes

5. Review Comments to the Author

Reviewer #1: Reviewer comments

This study asks an important question that will have implications for any analyses using OPC or OPCR. This is good, rigorous work, and I recognize that compiling and testing data collected in previous studies can be met with backlash from those involved in the original data collection. I want therefore to thank the authors for doing this study that will allow researchers moving forward to make informed decisions about which methods to use in their analyses.

There are two main topics I want to touch on related to this study: the first is the claim that OPC/R analyses are often done without any regard for phylogeny or phylogenetic signal. Of the 16 studies cited in Table 1, four appear to have no regard for phylogeny (cited as 46, 9, 51, and 10), and three account for phylogeny by only looking at crown taxa (e.g. bears (50), primates (29, 45), rodents (36)) as you suggest. All of the other studies take it into account either through statistical analyses (ANOVA, Mann-Whitney U test, pairwise phylogenetically informed regression analyses), or through their interpretation of their results (suggesting topographical analyses should be paired with phylogenetic analyses). It is well-established that phylogenetic signals are extremely strong in mammalian teeth, and so the claim that researchers have been conducting morphological studies without accounting for phylogeny must be backed up. Is their accounting for phylogeny insufficient? Can you demonstrate that?

The second thing I want to touch on is the issue of methodology between studies. Several times you mention that the methodology to determine OPC/R is variable between studies. This effectively means that researchers are each following their own methodologies and that these data cannot be comparable. This should be made quite clear; if people are not standardizing their dental meshes to 10 000 polygons, or are deviating from the original methods set out in Evans et al. 2007, then their data should absolutely not be combined.

64-68: It’s not just the tough cell structure of plants that causes differences in processing methods. Yes, vertebrate flesh is easier to break down, but it must be sliced with highly specialized dentition. A brief comment on this is warranted.

82: be specific when talking about diets. OPC and OPCR are used to infer the average diets that an animal has adapted to eat, as opposed to the specific diet of an individual.

87: Christison et al. 2022 is another example of this

88: I’d say “haphazardly” is unfair in this instance. Using published data is a very established method of conducting science, and before the present study it was a fair assumption that they would be transferrable.

92-93: see note about phylogeny

95: Reiterate the assumption at the beginning of the paragraph

96-97: the statement that “variation of these methods has been found to affect results” is vital to this study. If the data was collected by different methods, then they are not comparable. A more damning statement here would be to point out that researchers are varying in the way that they follow the methods, this is effectively new information since one would hope we were all following the methods correctly. More on this later (lines 324-332).

99-100: The statement that distinct dietary categories obscure dietary variation is correct; however, an explanation as to why people even use the categories and what they are based on is warranted. Typically, it is for analytical purposes, and again meant to reflect the average diet an animal has adapted to rather than the diet of the individual. Pineda-Munoz et al. 2017 reports how they established their categories based on stomach contents, for example. I’ve also used Elton Traits 1.0 (Wilman 2014), a database of species-level info from field guides, and dietary categories from Evans et al. 2007 and Van Valkenburgh et al. 2004.

Stable dietary categories are meant to work as general guidelines based on the average diets that an animal has adapted to eat. I am not defending them nor advocating them but merely explaining the reason they are used and can be useful especially when looking at extinct animals whose diets and ecologies are poorly-understood.

106-109: note on phylogeny

124: style preference: it can be useful to re-define acronyms the first time they are used in each section

169-174: For data extraction and comparing different dietary categories to each other, I wonder if it would be possible to standardize dietary categories. Studies should theoretically explain how they determined their dietary categories, and possibly lead to the data they used. For example if study A separated frugivores and folivores using a database of stomach contents, and study B only used a category of “herbivore” using a database of observed diets, it may be possible to look at the databases directly to determine dietary categories. If you’re able to eliminate the different ways diets are categorized as a variable it could strengthen your study.

251: phylogeny!

294-296: Nailing down the differences in methodology is vital to determining if these studies are comparable or not. I’d also like some clarification on OPC/R for tooth row vs individual teeth, because these data are not comparable. I know Pineda-Munoz et al 2017 uses average OPCR (OPCR divided by number of teeth in the tooth row) which would be more comparable to individual teeth, but still not perfect. It may be worth standardising all the data here to mean OPC/R if you want to ensure they are comparable.

You should also specify which individual teeth are being looked at; for example in carnivorans it is established that the carnassial teeth (m1 and P4) contain the most information about an animal’s diet, while some teeth may contain little to no dietary info. If individual teeth are being looked at but they are different teeth in the tooth row, this may significantly impact your results.

325-332: “similar enough methods” is unclear; how do they differ? Differences in methodology are crucial to figuring out if these data are comparable.

394: It is worth noting that thus far, no studies have examined the impact of mesowear on OPC/R values within individual species.

Citations

Christison, Brigid E., F. Gaidies, S. Pineda-Munoz, A. Evans, and D. Fraser. Dietary niches of creodonts and carnivorans of the late Eocene Cypress Hills Formation. Journal of Mammalogy 103: 2–17.

Evans A.R., Wilson G.P., Fortelius M., Jernvall J. 2007. High-level similarity of dentitions in carnivorans and rodents. Nature 445:78–81.

Pineda-Munoz S., Lazagabaster I.A., Alroy J., Evans A.R. 2017. Inferring diet from dental morphology in terrestrial mammals. Methods in Ecology and Evolution 8:481–491.

Van Valkenburgh B., Wang X., Damuth J. 2004. Cope’s rule, hypercarnivory, and extinction in North American canids. Science (New York, N.Y.) 306:101–104.

Wilman H., Belmaker J., Simpson J., de La Rosa C., Rivadeneira M., Jetz W. 2014. EltonTraits 1.0: species-level foraging attributes of the world’s birds and mammals. Ecology 95:2027.

6. PLOS authors have the option to publish the peer review history of their article (what does this mean?). If published, this will include your full peer review and any attached files.

Reviewer #1: **Yes: **Brigid E. Christison

---

## [Author Response · Author response to Decision Letter 0]

1 Sep 2023

We are grateful to the feedback offered for improving our submitted manuscript. Based on this feedback, we incorporated several new analyses that strengthened our conclusions and revealed valuable new insights.

In response to the request for an analysis of phylogenetic signal, we calculated Blomberg’s K for the dataset of one of the studies included in our meta-analysis. This analysis revealed that OPC(R) values are closely related to phylogeny and in some cases have a distribution that is not significantly different from a distribution due to a random walk along the phylogeny, supporting our overall claim about the importance of carefully considering the phylogenetic group of interest in OPC(R) analyses.

We repeated our analyses of variation among studies with a distinction between single- and multi-tooth studies, and were able to identify a trend for multi-tooth studies to find a greater difference in OPC(R) between herbivores and faunivores.

We followed the suggestion to emphasize the importance of differing methods among studies of OPC(R) by including a new table (Table 1) and reiterating this point in the text in both the Introduction and the Discussion.

Existing figures and tables were edited in accordance with the reviewer requests.

We appreciate the reviewer inquiries about our stance in regard to how phylogenetic signal has been dealt with previously. We agree that in most published studies phylogeny is properly accounted for either in the choice of sample or in formal analyses. We have clarified our manuscript to make more apparent that our concern is with comparing OPC(R) values across distantly related taxa, as is often done with fossils, and not with how phylogenetic signal is handled within most individual studies.

We also made the following changes, at the line numbers specified:

• Wording was changed to clarify the difference in digestability between plant and animal matter (61).

• The importance of differing methodologies was emphasized in the text (92).

• A table was added to highlight the varied and changing methods used in OPC(R) analysis (112).

• A table was added to clarify the diets seen in the included studies and how they were categorized

here (182).

• The two forest diagrams were relocated in the manuscript to be more relevant to the text (317,

323).

• Notes were added on intraspecific variation in OPC(R) seen in other studies (367).

• Table 4 was altered so that differences are now expressed as a correct “Percentage difference”

(374).

• Additional advice was added on using OPC(R) given the variation in methods currently seen and

strength of phylogenetic signal (428).

• Figures were changed to add clarity and distinguish single- from multi-tooth studies.

• The financial disclosure and funding information for the paper were brought into agreement.

• The manuscript and other submitted files were reformatted to match journal standards.

• Minor wording was changed throughout for readability.

• References were checked for retractions; no retractions were found.

• Three references were added: [68] Blomberg et al. 2003, [69] Revell et al 2012, and [71] Pampush et al. 2016

We thank you again for your feedback.

---

## [Decision Letter · Decision Letter 1]

20 Sep 2023

Dental complexity and diet in amniotes: a meta-analysis

PONE-D-23-17989R1

Dear Dr. DeMers,

We’re pleased to inform you that your manuscript has been judged scientifically suitable for publication and will be formally accepted for publication once it meets all outstanding technical requirements.

Kind regards,

Carlo Meloro

Academic Editor

PLOS ONE

Additional Editor Comments (optional):

The paper has been greatly improved and it is ready for publication I only recommend some clarification in the section of methods when you introduce the K statistic. It is good practise to describe how does it varies and what the variation of K means (e.g. K = 0 means data follow a star phylogeny while K = 1 data should follow variation expected by Brownian so on..). You also did an additional test to validate if your K values were different from what would be expected by BM. This again should be clarified since it is common knowledge that K=1 stands for Brownian Motion so I assume you used some additional scripts from Revell's blog.

Last thing, please provide better formatting for Table S2 that should go after Table S1 in the manuscript order (e.g. put the table in an horizontal rather than vertical page) and acknowledge also the reviewer effort in the acknowledgments section.

These are minor adjustments that I hope you can implement. You really did a good job in revising the paper.

Reviewers' comments:

Reviewer's Responses to Questions

**Comments to the Author**

1. If the authors have adequately addressed your comments raised in a previous round of review and you feel that this manuscript is now acceptable for publication, you may indicate that here to bypass the “Comments to the Author” section, enter your conflict of interest statement in the “Confidential to Editor” section, and submit your "Accept" recommendation.

Reviewer #1: All comments have been addressed

2. Is the manuscript technically sound, and do the data support the conclusions?

Reviewer #1: Yes

3. Has the statistical analysis been performed appropriately and rigorously? 

Reviewer #1: Yes

4. Have the authors made all data underlying the findings in their manuscript fully available?

Reviewer #1: Yes

5. Is the manuscript presented in an intelligible fashion and written in standard English?

Reviewer #1: Yes

6. Review Comments to the Author

Reviewer #1: The authors have responded to all of my comments regarding the manuscript. In particular, their clarifications about the previous use of OPC(R) in regards to taxonomic group help to strengthen their argument that the method should not be used more broadly. The added tables were a welcome addition as well that ensure transparency in methodology. This manuscript will prove extremely useful for those doing comparative analyses of tooth shape, especially for those attempting to infer diet from extinct taxa.

7. PLOS authors have the option to publish the peer review history of their article (what does this mean?). If published, this will include your full peer review and any attached files.

Reviewer #1: **Yes: **Brigid E Christison

---

## [Editor Report · Acceptance letter]

26 Sep 2023

PONE-D-23-17989R1 

Dental complexity and diet in amniotes: a meta-analysis 

Dear Dr. DeMers:

I'm pleased to inform you that your manuscript has been deemed suitable for publication in PLOS ONE. Congratulations! Your manuscript is now with our production department. 

Kind regards, 

on behalf of

Dr. Carlo Meloro 

Academic Editor

PLOS ONE